# Resting Energy Expenditure during Breastfeeding: Body Composition Analysis vs. Predictive Equations Based on Anthropometric Parameters

**DOI:** 10.3390/nu12051274

**Published:** 2020-04-30

**Authors:** Agnieszka Bzikowska-Jura, Adriana Szulińska, Dorota Szostak-Węgierek

**Affiliations:** Department of Clinical Dietetics, Faculty of Health Sciences, Medical University of Warsaw, E Ciolka Str. 27, 01-445 Warsaw, Poland; adriana.szulinska@gmail.com (A.S.); dorota.szostak-wegierek@wum.edu.pl (D.S.-W.)

**Keywords:** resting energy expenditure, predictive equations, breastfeeding, body composition, bioelectrical impedance analysis

## Abstract

Accurate estimation of energy expenditure in a breastfeeding woman is crucial for maintaining the proper nutritional status of the woman and healthy development of the infant. The current literature does not contain data regarding resting energy expenditure (REE) in breastfeeding women. Using mathematical equations is the most common method of REE assessment. However, due to changes in metabolism and body composition during pregnancy and lactation, the mathematical equations used among the general population may not apply. The aim of this study was to evaluate the resting energy expenditure of exclusively breastfeeding women by using body composition analysis–estimated REE (eREE) and to provide the most appropriate predictive equations–predicted REE (pREE) based on anthropometric parameters to estimate it. This was a pilot study with 40 exclusively breastfeeding women. Height and weight were measured and body composition analysis was performed. We predicted REE using fourteen self-selected equations, based on anthropometric parameters and/or age, and/or sex. The median eREE was 1515.0 ± 68.4 kcal (95% Cl, 1477–1582 kcal) and the pREE ranged from 1149.7 kcal (95% Cl, 1088.7–1215.0) by Bernstein et al., to 1576.8 kcal (95% Cl, 1479.9–1683.4), by Müller et al. Significant differences between eREE and all pREE were observed (*p* < 0.001, except Korth et al. equations). The Müller et al. equation was the most accurate with the smallest individual variation. All predictive equations showed low agreement, and in most cases, the results were underestimated. These findings indicate the need for further studies to propose more suitable methods to determine the energy requirements for breastfeeding women.

## 1. Introduction

Calorie intake corresponding to energy needs is necessary to maintain a healthy body weight, and in the case of young organisms also for their proper growth and development [1]. Breastfeeding is a period during which an adequate supply of energy and nutrients is particularly important. This is because women during pregnancy and in the postpartum period are particularly vulnerable to the occurrence of food deficiencies [2]. During this time, excessive weight gain often appears, which in the future may be associated with health problems as consequences of overweight and obesity [3]. An adequate consumption of energy and nutrients during lactation determines the proper nutrition status of a woman, facilitates weight control [4] and to some extent, affects the composition of the milk produced, which is important for proper development of the infant [5]. Therefore, in clinical practice, the most accurate assessment of energy demand possible during this period of life is crucial.

Resting energy expenditure (REE) is the energy expenditure of an individual who is not fasting and is the number of calories required for a 24 h period by the body during a non-active period. REE is the largest component (up to 70%) of total energy demand and can be measured using indirect calorimetry. Indirect calorimetry is a method that allows one to measure the basic energy expenditure based on the composition of the inhaled and exhaled air. Therefore, it lets one evaluate the actual energy cost of metabolic reactions taking place in the body. The use of indirect calorimetry is also possible in critically ill or mechanically ventilated patients, and allows for precise REE measurement and adjustment of energy supply to demand changing during illness, which is important for improving treatment outcomes. Indirect calorimetry is the gold standard for measuring REE and for guiding nutritional recommendations and support. However, several factors (e.g., agitation, fever, high positive end expiratory pressure—PEEP > 10 and high fraction of inspired oxygen—FiO2 > 80%) may limit the accuracy and/or the workability of the measurements [6]. Furthermore, there are some medical conditions that will exclude a subject from having the indirect calorimetry test conducted (e.g., suffering from claustrophobia, nausea, vomiting or lack of tolerance towards a face mask). Other limitations for performing indirect calorimetry in clinical practice are: equipment and maintenance costs, lack of trained staff, difficulties in interpreting the results and lack of time to carry out the measurements [7]. Therefore, indirect calorimetry is not available in many situations, and REE is often estimated using predictive equations [8]. However, there are some doubts as to whether the mathematical formulas developed for the general population can be used for lactating women. It must be stressed that women in the first few months after delivery have a higher percentage of body fat [9], which in turn is a factor directly affecting the amount of REE [8]. There is also some evidence that during lactation there may be a slight increase in basic energy requirements compared to the period before pregnancy. However, these data are not conclusive [10].

Since the most used equation, the Harris–Benedict equation in 1918 [11], nearly 200 published REE formulas have been published dealing with various conditions [12], and the body composition is relevant to assessing the validity of REE equations, which mainly depends on gender, age and weight [13].

In healthy subjects, fat free mass (FFM) is the most important determinant of REE and total daily energy expenditure (TEE) [14]. Wayer et al. [15] reported that FFM explained 72% of the variance in TEE and 66% of the variance in REE. Some studies have found that fat mass (FM) is an independent predictor of energy expenditure (EE) [15,16,17]. None of them have investigated the relationship between body composition and EE in breastfeeding women.

Bioelectrical impedance analysis (BIA) is a method that estimates the corporal compartments, including the quantity of liquid in the intracellular and extracellular spaces. BIA measurements are taken by injecting a small alternating current into the body. There are two main electrical properties that characterize body tissues, resistance and capacitance. Cell membranes conducting an electrical current behave similar to capacitors. Due to their ionic nature, body fluids are good conductors, while fat cells are not. Bone is also considered a non-conductor under typical BIA conditions. The lean mass is highly conductive due to the great quantity of water and electrolytes that it contains; therefore, it has low resistance. On the contrary, the fat mass, skin and bones are components of low conductivity, and therefore, of high resistance [18]. Assessment of REE using electrical bioimpedance is based on mathematical formulas that consider measured body composition parameters. This method is associated with lower costs than indirect calorimetry; its implementation is quick and simple; and the equipment is relatively light and easy to transport. However, certain rules must be followed to receive the correct results. Before proceeding to the measurement it is required to avoid food and drinks with caffeine for 4 h, intense physical activity for 12 h and diuretics for 6 h. [19]. Other factors that may distort the measurement results are severe obesity; water and electrolyte disturbances; and incorrect placement of the electrodes [20]. There are also contraindications regarding performing this test; e.g., pregnancy and the presence of a pacemaker [21].

Despite the fact that the bioelectrical impedance method is not considered as a gold standard in the assessment of resting energy expenditure, it can be applied in clinical practice [22]. Because this method uses information about FFM, which is more responsible for the amount of resting energy expenditure than body mass itself, it seems to be more useful than mathematical formulas based on anthropometric parameters [23]. 

The purpose of the present pilot study was to assess which of the available mathematic equations (pREE) best reflect the value of REE estimated by the method of electrical bioimpedance (eREE) in breastfeeding women, and thus which of them can be used to assess energy demand in this group.

## 2. Materials and Methods

### 2.1. Subjects and Data Collection

The study was carried out on breastfeeding women (*n* = 40) in their first month postpartum (3–4 weeks) in a maternity department in Warsaw. The inclusion criteria were: age ≥ 18 years, full-term delivery (gestational age ≥ 37 weeks) and exclusive breastfeeding. Exclusion criteria included: pre-existing chronic or gestational diseases and any contraindications that apply to body composition analysis (epilepsy, metal implants, pacemaker, defibrillator, stents, implanted devices which emit electronic signals). Body weight and height were measured using measurement station and column scales Seca 799 (±0.1 kg/cm). Body mass index (BMI) was calculated as the ratio between the body weight and the height squared (kg/m^2^). Interpretation of these results followed the international classification proposed by the World Health Organization (WHO): <18.5 kg/m^2^, underweight; 18.5–24.9 kg/m^2^, normal weight; 25.0–29.9 kg/m^2^, overweight; ≥30.0 kg/m^2^, obese [24]. The Ethics Committee of the Medical University of Warsaw (KB/172/115) approved the study protocol, and all the participating women gave written informed consent.

### 2.2. Bioelectrical Impedance Analysis

Whole body impedance (wrist to ankle) of women was measured using the Maltron BioScan 920-II multi-frequency bioelectrical impedance analyzer according to manufacturer’s instructions. Total body electrical impedance alternated with four different frequencies, 5, 50, 100 and 200 kHz. The subjects were measured in supine position, on a non-conductive surface, taking rest for about 3 min. Before placing the electrodes, the sites were cleaned using isopropyl alcohol, limiting the possible errors and to ensure the adherence. The whole-body impedance vector components, resistance (R) and reactance (Xc), were measured at the same time. On this basis, body fat, other components and REE were calculated. Before taking BIA measurements, the women were instructed according to Heyward and Stolarczyk (1996) [25] by the following guidelines: no heavy exercise 12 h before the test; no large meals or intake of caffeinated products 4 h before the test; consumption of liquids limited to 1% of body weight, or two 8 oz glasses of water, 2 h before the test.

### 2.3. Resting Energy Expenditure (REE) Predictive Equations

The predictive equations for resting energy expenditure (REE) used in our study were obtained by screening previous publications and are summarized in Table 1: Harris Benedict [11], Bernstein et al. [26], Owen et al. [27], Mifflin et al. [28], Schofield [29], FAO/WHO (including only weight, as well as weight and height) [30], Institute of Medicine [31], Müller et al. [32], Korth et al. [23], De Lorenzo et al. [33], Lazzer et al. [34], Henry [35] and Huang et al. [36]. We selected REE predictive equations dedicated to adults, based on the following criteria: body weight, height, sex and age. The REE was predicted for each equation separately with the actual body weight and height at the time of body composition measurements.

### 2.4. Statistical Analysis

Results are presented as means ± standard deviations, medians and interquartile ranges. The difference between the estimated and predicted REE (ΔREE) is expressed as an absolute value (kcal/day, mean bias) and percentage (%, relative bias) [37]. Relative bias (%) was calculated as follows: (ΔREE _mean bias_)/REE _estimated_ × 100. A measurement was considered inaccurate when the relative bias was greater than ±10% of the estimated REE, and the number of subjects with an inaccurate prediction was calculated [38]. Pearson’s correlation analysis was performed to evaluate correlations between weight and body composition parameters. Correlation between pREE and eREE was estimated with Spearman correlation coefficient. A Bland–Altman plot analysis was conducted to examine the agreement between the measured and estimated REE. The paired *t* test was used to examine the mean difference between the estimated and predicted REE. All analyses were performed using Statistica 12PL, Tulusa, USA, and IBM Statistics 21 New York, NY, USA. A *p*-value below 0.05 was adopted as statistically significant.

## 3. Results

### 3.1. Subjects’ Characteristics and Body Composition Parameters

For this study we assessed 41 exclusively breastfeeding women. After data consistency analysis, we excluded those with an eREE < 500 kcal (*n* = 1), resulting in a final sample of 40 participants. The descriptive characteristics of the study population are shown in Table 2. BMI values ranged from 16.46 to 32.11 kg/m^2^ and age ranged from 24 to 43 years. Regarding nutritional status, assessed with BMI, it was found that 67.5% (*n* = 27) of the sample had normal body mass, 20% (*n* = 8) was overweight, 7.5% (*n* = 3) was underweight and 5% (*n* = 2) was obese. We observed a statistically significant positive correlation between body weight and percentage of fat mass (r = 0.92; *p* = 0.000), and a negative correlation with the percentage of total body water (r = −0.84; *p* = 0.000).

### 3.2. Estimated and Predicted Resting Energy Expenditure (REE)

The median eREE was 1515.0 ± 68.4 kcal (95% Cl, 1477–1582 kcal) and the median pREE ranged from 1149.7 kcal (95% Cl, 1088.7–1215.0) by Bernstein et al., to 1576.8 kcal (95% Cl, 1479.9–1683.4) by Müller et al. Using the Wilcoxon test, we observed significant differences between eREE and all pREE. For all formulas except Korth et al., *p* < 0.001. The detailed results of characterization and the results of correlation between eREE and pREE are described in Table 3. All equations showed significant results for the Spearman correlation. The correlations observed were classified as strong, and the best results were obtained by Miffilin et al. (r = 0.872; *p* < 0.001) and Korth et al. (r = 0.870; *p* < 0.001).

Table 4 shows the absolute difference between the predicted and estimated REE in each equation. The mean estimated bias was smaller in the equations by Korth et al. and Müller et al. than in other equations (<|67 kcal/day|). The range of the 95% limits of agreement (LoA) was smaller in the equations by IOM and Harris–Benedict et al. than in other equations (95% confidence interval of LoA, <330 kcal/day), although all equations showed wide LoA. The Müller et al. equation was the most accurate with the smallest individual variation. The relative bias is shown in Figure 1. Three equations were characterized by a prediction within ±10% of the measured REE in ≥80% of subjects: Müller et al., Harris–Benedict and Korth et al. (Figure 2).

The correlation between eREE and pREE using the Bland-Altman method is shown in Figure 3. All the equations presented high dispersion of the points in the graph, which means that, according to Bland-Altman method, all the equations presented low consistency with the body composition analysis in our study group of breastfeeding women. That which showed less dispersion was Owen et al.

## 4. Discussion

The present study compared REE estimated by BIA with REE estimated from selected predictive equations, based on anthropometric measurements and/or sex and/or age. Many studies have focused on the effects of clinical conditions on the REE; to the best of our knowledge, however, no studies have addressed the validity of predictive equations for exclusively breastfeeding women. Proper supply of energy and nutrients during lactation is crucial, both for a woman and her child. First of all, it prevents a woman’s nutritional deficiencies and their health consequences, and it also allows her to return to pre-pregnancy weight in a safe and healthy way. Nutrition of a nursing woman can also affect the composition of milk and the lactation process itself, which is crucial for proper infant development. Meanwhile, many studies confirm that the intake of energy and nutrients by breastfeeding women usually does not meet the general recommendations in this group [40,41,42,43]. The low accuracy of prediction equations for evaluating REE in exclusively breastfeeding women implies greater difficulty in establishing the proper nutritional interventions for this population. It should be stressed that the various changes that occur in a woman’s body after childbirth make her energy requirement very specific. Therefore, it is essential to know the most reliable method to estimate it [44].

So far, predictive equations remain the most common REE estimation method. They allow a rapid calculation of REE using basic anthropometric data (weight and height) and have been validated among different population groups. Most of these equations have been developed in healthy subjects, resulting in large errors in case of chronic diseases or different physiological status, such as pregnancy or lactation despite, the use of the correction factors [6]. Other methods for assessing REE have been explored and compared to indirect calorimetry in order to find a valid alternative. One of these methods is BIA, based on body composition analysis. This approach has been shown to be quite inaccurate in clinical populations compared to indirect calorimetry and cannot be adopted in critically ill patients [45,46,47] due to their abnormalities in hydration state and serum electrolyte concentrations that cause errors in the BIA-derived estimates of FFM and FM [14]. In our population those abnormalities were excluded, so possible errors were minimized.

The agreement of predictive equations compared to indirect calorimetry is low, and does not exceed 55%, especially in critically ill patients and those with extreme BMI (BMI < 16 kg/m^2^ and BMI > 40 kg/m^2^) [48,49]. Predictive equations tend to overestimate (e.g., chronic kidney disease [50], cancer [51]) or underestimate (e.g., chronic obstructive pulmonary disease [52], diabetes type 2 [53] REE in patients with chronic diseases). Excluding critically ill patients, differences and errors are mainly due to an excessive or deficient FM, which is less metabolically active than the FFM, and to the body weight considered for the calculation (current, ideal, or estimated) [45]. Although the nutritional status of our participants was varied, BMI values ranged only from 16.46 to 32.11 kg/m^2^.

This study found low agreement of all predictive equations to estimate women’s REE in the exclusively breastfeeding period, and, in all cases except the Müller et al. equation, the results were underestimated. This seems to confirm the hypothesis that during lactation a slight increase in REE may occur [10]. However, among the evaluated formulas, Korth et al., and Müller et al. were the better predictors of REE for this population (ΔREE - 51.37; −66.43 kcal/day, respectively). However, the Müller et al. equation had the highest level of accuracy at an individual level, and therefore this may be the best equation for predicting REE of exclusively breastfeeding women in clinical practice.

According to our knowledge, the only study assessing the compliance of pREE and measured REE conducted in a similar group seems to be the study performed by de Sousa et al. [44]. The study involved women in the immediate postpartum period (up to 10 days after delivery). REE was measured by indirect calorimetry (mREE) and predicted using eight equations. Using the Wilcoxon signed rank test, the authors found that the best predictor of REE was the Harris–Benedict equation, with lower difference (*p* = 0.876), better median of adequacy (99.8%), and better interclass correlation coefficient (0.289). The Schofield formula was next, with greater percentage of accuracy (33.3%) and lower opposite agreement. The smallest dispersion in the Bland-Altman test were obtained for the results of Harris–Benedict and Owen equations, which is consistent with our results. The authors of current study emphasize that none of the formulas given are sufficiently accurate to be used in this group for REE assessment. It should again be noted that there is lack of other studies comparing REE estimated with BIA and pREE. Barak et al. [14] used BIA to predict REE in hospitalized patients receiving nutrition support. They found that BIA-derived body composition estimates may be used to more accurately predict the energy requirements than calculations based on mathematical formulas. However, they emphasized that one of the limitations of their study was the use of a single-frequency (50 kHz) device. What is more, the authors indicated that the use of multi-frequency bioelectrical impedance, which was used in our study, would be more accurate and was advised. There are studies assessing the suitability of mathematical formulas, including data obtained from BIA. However, such formulas do not prove to be more accurate than those based on anthropometric measurements. Both are characterized by an accuracy of less than 57% compared to the results obtained with indirect calorimetry [54].

In the study of Pereira et al. [22], whole body calorimetry measurements were performed and pREE was calculated using mathematical formulas among women three months after delivery and later nine months after delivery, both lactating and non-lactating. They found that the best equation predicting REE was the dietary reference intake equation at three months postpartum (−7 kcal, −0.1%; absolute and percentage bias, respectively), and the Harris–Benedict equation at nine months postpartum (−17 kcal, −0.5%). At an individual level, the FAO/WHO height and weight equation was the most accurate at three months postpartum (100% accuracy) and nine months postpartum (98% accuracy), with the smallest limits of agreement (LoA). However, there are indications that the selection of mathematical formulas for REE assessment may depend on individual body weight. 

This hypothesis is confirmed by the results of the research of Amaro-Gahete et al. [55], suggesting the FAO/WHO equation proved to be the most suitable for people with normal body weight, whereas for overweight and obese people the best were the Livingston and Owen formulae, respectively. Weijs et al. [38] came to similar conclusions. They found that for women with normal body mass, the Huang equation was the most accurate. Its effectiveness, however, definitely decreases with BMI > 45 kg/m^2^. The most accurate in this group proved to be the Siervo equation, while the FAO/WHO and Schofield equations should not be applied at BMI > 45 kg/m^2^. Nevertheless, according to researchers, the Harris–Benedict and Mifflin equations can still be successfully used in REE assessment regardless of the BMI value. However, such a relationship does not appear in the Frankenfield et al. study [49], where among the equations that are the most common in clinical practice for subjects with normal body weight and with obesity, the Mifflin-St Jeor equation proved to be the most reliable. For all tested equations (Harris–Benedict, Mifflin, Owen, FAO/WHO) the accuracy decreased as the BMI of the subjects increased. However, in the case of the Mifflin formula, the decrease was the smallest. Predicted accuracy for normal body weight was 82%, and for obesity it was 70%. Wilms et al. [56], who investigated only women with varying degrees of obesity, found that none of the mathematical formulas used, either based on anthropometric data or based on body composition parameters, should be used to assess REE in this group. In this study, with increase in mREE, the discrepancies between pREE and mREE also increased. Predicted accuracy was not greater than 70% for any of the examined equations, and for Bernstein and Owen formulas it was 7% and 20%, respectively. It was also noted that the differences between mREE and pREE are not dependent on percentage fat mass.

It was also observed [22] that REE prediction error and individual accuracy were not improved with the inclusion of body composition variables, contrary to expected patterns, because FFM is a major determinant of REE [57]. FFM is comprised of tissues and organs with different metabolic rates, ranging from 13 kcal/kg for skeletal muscle to 440 kcal/kg for heart and kidneys [58]. Thus, small differences in organ size can significantly affect REE, which might not be captured when FFM and FM are used in predictive equations. In our study, we used an analyzer assessing not only FFM and FM in total, but also other parameters, such as BCM (body cell mass). BCM is the metabolically active cell mass involved in O_2_ consumption, CO_2_ production and energy expenditure; thus, its measurement has been suggested as a tool for the evaluation of nutritional status. Since BCM is closely related to energy expenditure, it could also represent a good reference value for the calculation of nutrient needs [59]. 

The strengths of this study are the use of advanced techniques to assess maternal body composition in accordance with the recommended protocol [25], which allowed possible errors in body composition to be minimized. A basic limitation of our study was the lack of indirect calorimetry, which is considered as the gold standard with which to measure REE. The other limitations of this study involved convenience sampling, the modest number of participants and their different nutritional statuses. Further, the observations made in this investigation are specific to our Caucasian populations and should not be generalized to other ethnic groups. All indicated limitations decreased the representativeness of the study, and extrapolation of results should be performed with caution.

This study demonstrates a wide variation in accuracy for REE predictive equations in exclusively breastfeeding women. The highest level of accuracy at an individual level was gained by the Müller et al. equation; thus, we recommend it for predicting REE of exclusively breastfeeding women in clinical practice. The equipment for BIA is far less costly and easier to use than an indirect calorimetry machine and could be easily available to nutrition support personnel (e.g., dietitians) in all healthcare facilities. Further, investigations of a larger exclusively breastfeeding female populations, and a comparison with a control group (non-breastfeeding women after delivery) should be undertaken to confirm the use of multifrequency BIA.

## Figures and Tables

**Figure 1 nutrients-12-01274-f001:**
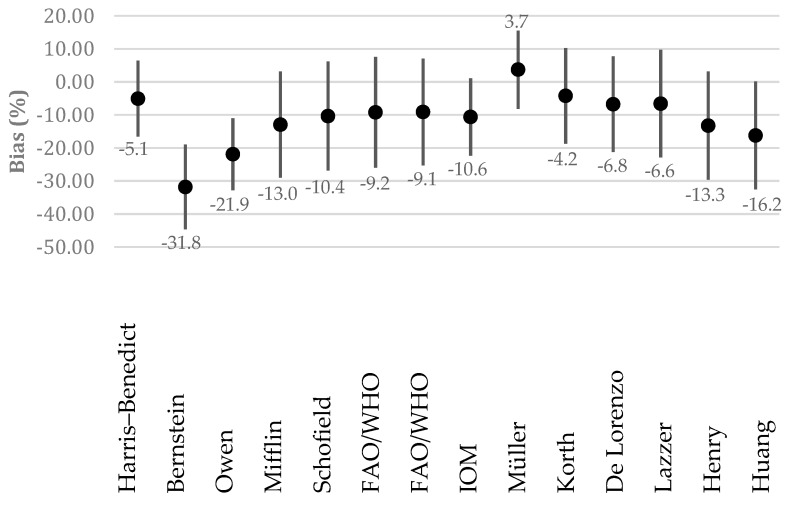
Percent bias of equations for the predicted REE compared with estimated REE (mean ± standard deviation).

**Figure 2 nutrients-12-01274-f002:**
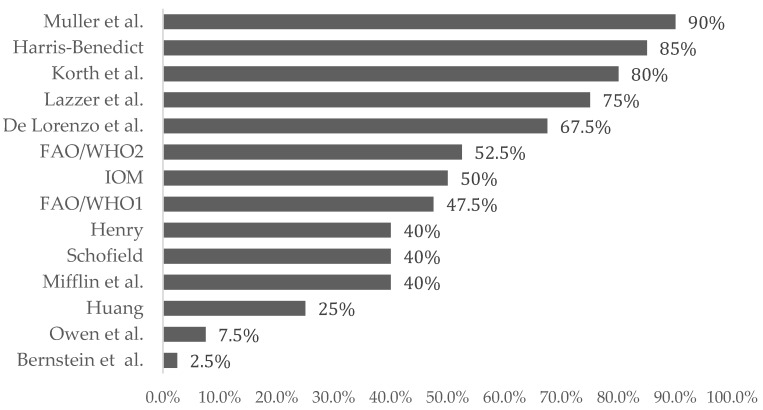
Percent of REE estimates with an individual bias (%) within ±10% of the estimated REE for each prediction equation. FAO/WHO1—only weight. FAO/WHO2—including weight and height.

**Figure 3 nutrients-12-01274-f003:**
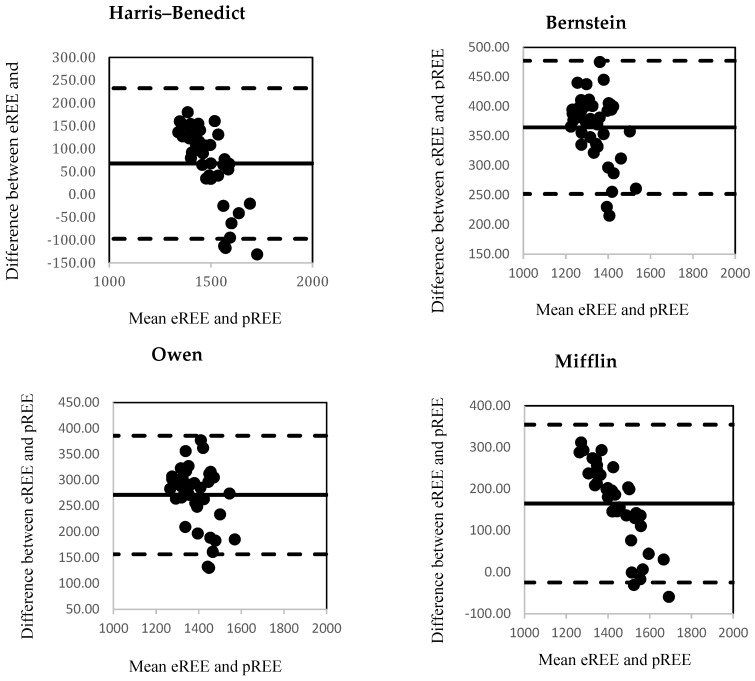
Individual consistency by the Bland-Altman method. ^1^ FAO/WHO—for weight only. ^2^ FAO/WHO—for weight and height.

**Table 1 nutrients-12-01274-t001:** Resting energy expenditure (REE) predictive equations.

Equations	Factors Used for Calculation	REE Predictive Equations (kcal/d)
Harris Benedict	Sex, W (kg), H (cm), age (y)	W × 9.5634 + H × 1.8496 − age × 4.6756 + 655.0955
Bernstein et al.	Sex, W (kg), H (cm), age (y)	7.48 × W − 0.42 × H − 3 × age + 844
Owen et al.	Sex, W (kg)	W × 7.18 + 795
Mifflin et al.	Sex, W (kg), H (cm), age (y)	9.99 × W + 6.2 × H − 4.92 × age − 161
Schofield	Sex, W (kg), H (m), age (y)	Age 18–30 y: (0.057 × W + 1.148 × H + 0.411) × 239 Age 31–60 y: (0.034 × W + 0.006 × H + 3.53) × 239
FAO ^1^/WHO ^2^	W (kg)	Age 18–30 y: 16.7 × W + 496 Age 31–60 y: 8.7 × W + 829
FAO/WHO	W (kg), H (m)	Age 18–30 y: 13.3 × W + 334 × H + 35 Age 31–60 y: 8.7 × W − 25 × H + 865
IOM ^3^	W (kg), H (m), age (y)	247 − 2.637 × age + 401 × H (m) + 8.6 × W
Müller et al.	Sex, W (kg), age (y)	(0.047 × W + 0.01452 × age + 3.21) × 239
Korth et al.	Sex, W (kg), H (cm), age (y)	(41.4 × W + 35 × H − 19.1 × age − 1731.2)/4.186
De Lorenzo et al.	Sex, W (kg), H (cm), age (y)	(46.322 × W + 15.744 × H − 16.66 × age + 944)/4.186
Lazzer et al.	Sex, W (kg), H (m), age (y)	(0.042 × W + 3.619 × H − 2.678) × 239
Henry	Sex, W (kg), age (y)	Age 18–30 y: (0.0546 × W + 2.33) × 239 Age 31–60 y: (0.0407 × W + 2.9) × 239
Huang et al.	Sex, W (kg), H (cm), age (y)	10.158 × W + 3993 × H − 1.44 × age + 60.655

W—weight; H—height; ^1^ FAO—Food Agriculture Organization; ^2^ WHO—World Health Organization; ^3^ Institute of Medicine.

**Table 2 nutrients-12-01274-t002:** Subjects’ anthropometric data and body composition measures.

	Mean ± SD	Median (Interquartile Range)
**Age (years)**	32.1 (6.2)	31.0 (30.0–35.0)
Height (cm)	166.6 (6.6)	166.5 (162.0–172.5)
**Pre-pregnancy weight**	61.4 (10.8)	58.0 (53.8–69.0)
Pre-pregnancy body mass index (kg/m^2^)	22.1 (3.3)	21.1 (19.5–23.7)
**Weight gain during pregnancy**	14.5 (4.6)	14.0 (11.5–16.5)
Weight at first month postpartum (kg)	64.5 (12.2)	62.3 (54.8–70.9)
Body mass index at first month postpartum (kg/m^2^)	23.0 (3.6)	22.7 (20.4–24.8)
Fat mass–FM (kg)	19.8 (10.3)	17.9 (11.3–23.0)
Fat mass–FM (%)	28.2 (8.4)	28.5 (20.6–33.0)
Fat free mass–FFM (kg)	45.4 (3.9)	45.7 (43.0–48.4)
Fat free mass–FFM (%)	71.8 (8.4)	71.5 (67.0–79.4)
Total body water–TBW (L)	32.4 (3.8)	31.2 (29.4–35.2)
Total body water–TBW (%)	51.2 (5.1)	50.3 (47.0–55.3)
Extracellular water–ECW (L)	15.0 (1.9)	14.7 (13.8–16.3)
Extracellular water–ECW (%)	46.3 (2.7)	46.4 (45.4–48.0)
Intracellular water–ICW (L)	17.4 (2.3)	16.8 (16.0–18.9)
Intracellular water–ICW (%)	53.7 (2.7)	53.6 (52.0–54.7)
ECW/ICW	0.87 (0.09)	0.87 (0.83–0.92)
Body cell mass BCM (kg)	23.9 (2.9)	23.6 (22.3–25.9)
Extracellular mass–ECM (kg)	21.5 (1.9)	21.4 (20.1–23.3)
Protein mass–PM (kg)	9.0 (1.4)	9.0 (8.5–9.9)
Muscles (kg)	19.9 (1.9)	19.8 (18.8–21.4)
Minerals (kg)	3.8 (0.6)	3.7 (3.5–4.1)
Total body potassium–TBK (g)	106.4 (12.0)	104.6 (98.7–114.8)
Total body calcium–TBCa (g)	892.3 (87.2)	879 (836.5–953)
Glycogen (g)	415.7 (38.1)	418.5 (391.0–444.0)
Dry weight (kg)	63.6 (12.2)	61.2 (53.5–69.9)
Body volume (L)	62.3 (13.1)	59.8 (51.5–69.5)

**Table 3 nutrients-12-01274-t003:** Characterization of the eREE and pREE.

Method	Energy Expenditure (kcal/day)		
	Median	95% Cl	Spearman Correlation Coefficient
BIA ^1^	1515.0 ± 68.4	1477.0–1582.0	-
Harris–Benedict	1441.0 ± 131.2	1361.5–1551.1	0.854 *
Bernstein et al.	1149.7 ± 92.0	1088.7–1215.0	0.818 *
Owen et al.	1236.6 ± 90.2	1179.1–1303.7	0.797 *
Mifflin et al.	1344.9 ± 150.3	1230.1–1476.9	0.872 *
Schofield	1366.2 ±159.7	1276.8–1495.3	0.820 *
FAO/WHO ^2^	1381.5 ± 162.8	1290.0–1516.1	0.791 *
FAO/WHO ^3^	1383.8 ±159.7	1294.2–1518.1	0.825 *
IOM	1375.0 ± 123.1	1280.7–1480.3	0.866 *
Müller et al.	1576.8 ± 143.8	1479.9–1683.4	0.748 *
Korth et al.	1461.8 ± 159.7	1339.5–1601.1	0.870 *
De Lorenzo et al.	1423.2 ± 152.5	1316.1–1552.6	0.858 *
Lazzer et al.	1423.8 ± 163.1	1298.3–1576.5	0.850 *
Henry	1328.0 ± 149.7	1248.5–1461.7	0.793 *
Huang et al.	1309.9 ± 142.2	1208.1–1427.4	0.841 *

^1^ Bioelectrical impedance analysis; ^2^ including only weight; ^3^ including weight and height; r < 0.30 weak linear correlation; r = 0.31–0.59 moderate linear correlation; * *p* < 0.05; r = 0.60–0.89 strong linear correlation; r = 0.90–1.00 highly strong linear correlation [39].

**Table 4 nutrients-12-01274-t004:** Means and standard deviations (SD) of mean differences of the estimated and predicted REE.

	ΔREE ^3^ kcal/day	SD	ΔREE ^1^ + 1.96 SD	ΔREE ^1^−1.96 SD	Range of LoA ^4^
Harris–Benedict	67.54 ***	84.10	−97.30	232.37	329.67
Bernstein et. al.	364.54 ***	57.56	251.73	477.35	729.08
Owen et al.	271.13 ***	58.41	156.64	385.62	542.25
Mifflin et al.	164.86 ***	96.70	−24.68	354.40	379.08
Schofield	131.91 ***	111.66	−86.93	350.76	437.70
FAO/WHO ^1^	117.23 ***	116.84	−111.77	346.23	458.00
FAO/WHO ^2^	116.40 ***	111.38	−101.90	334.69	436.59
IOM	140.86 ***	74.27	−4.71	286.43	291.14
Müller et al.	−66.43 **	104.99	−272.22	139.36	411.58
Korth et al.	51.37 *	104.49	−153.42	256.17	409.58
De Lorenzo et al.	86.88 ***	101.95	−112.94	286.70	399.65
Lazzer et al.	82.51 ***	110.23	−133.55	298.57	432.11
Henry	168.60 ***	105.29	−37.76	374.97	412.73
Huang et al.	203.76 ***	93.32	20.85	386.67	407.52

^1^ Including only weight; ^2^ including weight and height; ^3^ ΔREE—mean difference between estimated and predicted REE; ^4^ LoA—limit of agreement. Paired t test was performed to examine the mean difference between estimated and predicted REE. * *p* < 0.05; ** *p* < 0.001; *** *p* < 0.00.

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
