# Peer review of "Resting Energy Expenditure during Breastfeeding: Body Composition Analysis vs. Predictive Equations Based on Anthropometric Parameters"

_nutrients, 2020, doi:10.3390/nu12051274_

Round 1

Reviewer 1 Report

I would thanks the authors of this paper to have reconsidered this argument, which remains controversial but reinforce the new vision that the breastfeeding is very important for both mother and new born. 

During pregnancy, a female’s body undergoes many dramatic changes in metabolism that may predispose her to certain health disparities, if not reversed. When breastfeeding is initiated, a metabolic shift occurs that alters resource allocation from storage to milk synthesis. Indeed, breastfeeding results in improved glucose handling via a decrease in insulin production, improved insulin sensitivity, and a drop in β-cell proliferation. In addition, lipid metabolism is decreased in metabolically active tissues and lipid stores are mobilized to facilitate lipid transport to the mammary gland for milk synthesis. With that said I see that the main  strengths of this study are surely  the use of advanced techniques to assess maternal body
composition, which allowed possible errors in body composition to be minimized, but the great limitations is the evaluation of different nutritional status and the use of a control population such as the bottle-fed women.

These for future studies is strongly advised.

Author Response

Dear Reviewer,

we would like to thank for your review and comments. As we outlined in the summary, one of the limitations of our study was the varied nutritional status of breastfeeding women. In the future, we are planning to expand the size of the study group, which will allow the division based on the nutritional status. We agree with your feedback, that for the future study the use of a control population should be considered and crucial  for more valuable conclusions – we added this comment in the summary (lines 382-383).

Thank you for your time and consideration. We look forward to hearing from you.

Sincerely yours,

Agnieszka Bzikowska-Jura

Adriana Szulińska

Dorota Szostak-Węgierek

Reviewer 2 Report

The study by Bzikowska-Jura assesses which mathematic equation best reflects the value of resting energy expenditure (REE) estimated by electrical bioimpedance (eREE). The study is relevant and useful, but one important issue needs to be adressed:

The authors state correctly that REE can be measured by using indirect calorimetry which is probably the gold standard way of measuring REE. Although bioelectrical impedance analysis is far more precise that an estimate of REE based on mathematic equations there are some limitations using the BIA compared to indirect calorimetry. An elaboration on the two methods and the strengths and limitations of the methods should be added to the introduction and discussion sections.

Author Response

Dear Reviewer,

we would like to thank the reviewer for comments and suggestions that will make this paper more valuable for readers. We take your concerns seriously and have addressed them to the best of our abilities. We added information concerning strengths and limitations of indirect calorimetry and BIA in the introduction (lines 49-62; 85-93) and discussion (lines 302-312) sections.

Thank you for your time and consideration. We look forward to hearing from you.

Sincerely yours,

Agnieszka Bzikowska-Jura

Adriana Szulińska

Dorota Szostak-Węgierek